# Use of dual-flow bioreactor to develop a simplified model of nervous-cardiovascular systems crosstalk: A preliminary assessment

Nicoletta Marchesi[1☉], Annalisa Barbieri[1☉], Foroogh Fahmideh[1], Stefano Govoni[1], Alice Ghidoni[2], Gianfranco Parati[3,4], Emilio Vanoli[5,6], Alessia Pascale[1‡]*, Laura Calvillo[3‡]*

1 Department of Drug Sciences, Pharmacology Section, University of Pavia, Pavia, Italy, 2 Center for Cardiac Arrhythmias of Genetic Origin, Istituto Auxologico Italiano, IRCCS, Milan, Italy, 3 Department of Cardiovascular, Neural and Metabolic Sciences, IRCCS Istituto Auxologico Italiano, Milan, Italy, 4 Department of Medicine and Surgery, University of Milano-Bicocca, Milano, Italy, 5 Department of Molecular Medicine, University of Pavia, Pavia, Italy, 6 Cardiovascular Department, IRCCS Multimedica, Sesto San Giovanni, Italy

☉ These authors contributed equally to this work.
‡ These authors also contributed equally to this work.
* l.calvillo@auxologico.it (LC); alessia.pascale@unipv.it (AP)

**Data Availability Statement:** All relevant data are within the manuscript. Supplement data (S1 raw images) and bioreactor sterilizing protocol are held

## Abstract

Chronic conditions requiring long-term rehabilitation therapies, such as hypertension, stroke, or cancer, involve complex interactions between various systems/organs of the body and mutual influences, thus implicating a multiorgan approach. The dual-flow IVTech Live-Box2 bioreactor is a recently developed inter-connected dynamic cell culture model able to mimic organ crosstalk, since cells belonging to different organs can be connected and grown under flow conditions in a more physiological environment. This study aims to setup for the first time a 2-way connected culture of human neuroblastoma cells, SH-SY5Y, and Human Coronary Artery Smooth Muscle Cells, HCASMC through a dual-flow IVTech Live-Box2 bioreactor, in order to represent a simplified model of nervous-cardiovascular systems crosstalk, possibly relevant for the above-mentioned diseases. The system was tested by treating the cells with 10nM angiotensin II (AngII) inducing PKCβII/HuR/VEGF pathway activation, since AngII and PKCβII/HuR/VEGF pathway are relevant in cardiovascular and neuroscience research. Three different conditions were applied: 1- HCASMC and SH-SY5Y separately seeded in petri dishes (static condition); 2- the two cell lines separately seeded under flow (dynamic condition); 3- the two lines, seeded in dynamic conditions, connected, each maintaining its own medium, with a membrane as interface for biohumoral changes between the two mediums, and then treated. We detected that only in condition 3 there was a synergic AngII-dependent VEGF production in SH-SY5Y cells coupled to an AngII-dependent PKCβII/HuR/VEGF pathway activation in HCASMC, consistent with the observed physiological response *in vivo*. HCASMC response to AngII seems therefore to be generated by/derived from the reciprocal cell crosstalk under the dynamic inter-connection ensured by the dual flow LiveBox 2 bioreactor. This system can represent a useful tool for studying the crosstalk between organs, helpful for instance in rehabilitation research or

in the public repository: https://doi.org/10.5281/
zenodo.4106227 https://doi.org/10.5281/zenodo.
4246966.

**Funding:** This work was financially supported by
Association I-CARE Europe Onlus. The funders had
no role in study design, data collection and
analysis, decision to publish, or preparation of the
manuscript.

**Competing interests:** The authors have declared
that no competing interests exist.

when investigating chronic diseases; further, it offers the advantageous opportunity of culti-
vating each cell line in its own medium, thus mimicking, at least in part, distinct tissue *milieu*.

## Introduction

Biomedical molecular research aimed to the study of complex relationships between various tis-
sues, as it happens in chronic diseases or when investigating resilience from a traumatic event,
such as for example in rehabilitation or during aging, needs translationally relevant experimental
models. *In vivo/ex vivo* models using laboratory animals or classic *in vitro* cultures on petri
dishes have been extensively employed so far allowing researchers to achieve many important
findings. However, cells growing in a petri dish do not behave like the original cells belonging to
organs in living organisms. They are not connected with the whole complex environment and,
furthermore, they are seeded on a hard matrix without being subjected to flow conditions [1].
Overall, this setup is far from properly reproducing the physio-mechanical characteristics of the
organ of interest, and cells do not communicate with other different cell types neither bioelectri-
cally nor through biohumoral exchange. Of note, though, a static system is useful to study reac-
tions to specific stimuli, at both biochemical and molecular level. On the other side, such a
system is not representative of the exchange of information through various mechanisms occur-
ring physiologically between the various cell types located in different organs. These complex
interactions are better modelled in *in vivo* models that, on the other hand, may be hard to study
at molecular level. The recent development of Next-Generation In Vitro Testing Tools, engi-
neered within the 3Rs research, opens a new scenario to explore living systems providing a
bridge between the use of cultured cells on a petri dish and *in vivo/ex vivo* experiments on small
laboratory animals. An example of such a tool are IVTech Bioreactors [1], where, within the tis-
sue engineering field, a bioreactor is defined as a device able to simulate a physiological environ-
ment allowing cell or tissue growth. Inside IVTech Bioreactors, cells are subjected to the
physiological shear stress and nutrient absorption typical of the blood stream [1, 2], and two cell
types can be connected to study their reciprocal crosstalk under physiological or pathological
conditions. This system is modular, with culture chambers designed to be added sequentially or
in parallel, thus simulating a multiple organ system [1, 3–5]. Further, it is designed to be consis-
tent with plates or transwells, thus allowing the use of standard protocols for *in vitro* procedures.

The primary goal of the present study was to create for the first time an *in vitro* model able to
connect two distinct cell types, namely human neuroblastoma cells (SH-SY5Y; widely used as a
neuronal-like cellular model [6]) and Human Coronary Artery Smooth Muscle Cells
(HCASMC), in a dual-flow IVTech LiveBox2, thus representing a simplified model of the ner-
vous-cardiovascular systems crosstalk. Since bioreactor technology is relatively recent, informa-
tion on a number of methodological aspects is still lacking (i.e. culture conditions, growing
condition for several types of cells, flow parameters etc.). Moreover, with respect to the classic *in
vitro* models, extensively used within the last decades and for which several procedures are avail-
able, only a few tried-and-tested experimental protocols are available for researchers to work
under dynamic conditions, especially when considering specific interconnected cell lines, as
SH-SY5Y cells and HCASMC. Therefore, our primary aim was to develop and share with the sci-
entific community a new co-culture set-up involving specifically SH-SY5Y cells and HCASMC,
where each cell type is seeded in its own medium, to avoid a forced adaptation to a different cul-
ture medium, and under flow conditions, thus mimicking a more physiological environment.

Secondly, we aimed to explore some aspects of the nervous-cardiovascular systems cross-
talk. The dynamic reciprocal relationship between brain and heart is important both in acute,

second by second regulation, and in chronic derailment situations when either organ is suffering because of a disease or of an improper drug use (see [7] for an extensive analysis of these relationships). In particular, within this context, we chose to explore some of the relationships involving AngII treatment effects on the two cell types, seeded inside IVTech Bioreactor Live-Box 2 and connected under flow conditions, and to study AngII-dependent PKCβII/HuR/VEGF (vascular endothelial growth factor) cascade activation in different experimental settings.

## Material and methods

### IVTech LiveFlow® and LiveBox2

The system (IVTech Srl., Massarosa, LU, Italy) consisted of a peristaltic pump (IVTech Live-Flow®), which creates the flow, connected with modular and transparent double flow bioreactors named LiveBox2 (LB2), where cells are seeded. Cell medium, in the supplied 25 ml plastic bottle, is connected to the IVTech LiveFlow® and to the LB2 by silicon tubes. LB2 is a dual-flow IVTech bioreactor formed by two chambers, upper and lower, developed to model physiological barriers *in vitro* (Fig 1). In particular, LB2 consists of three parts (Fig 1C):

1. an apical chamber with a wet volume of 1.5 mL, equipped with an inlet and an outlet tube;

2. a basal chamber with a wet volume of 1 mL, equipped with an inlet and an outlet tube;

3. a membrane holder, placed between the two chambers.

All the components of the IVTech bioreactor were autoclaved and the entire experiment was performed under a laminar flow hood (Fig 1B). Membranes were conditioned keeping them in ethanol 70% for two hours and exposed to UV light for 15 min, before cell seeding.

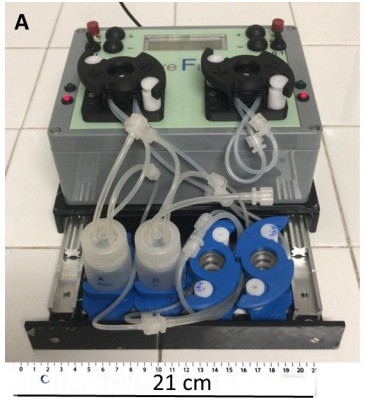
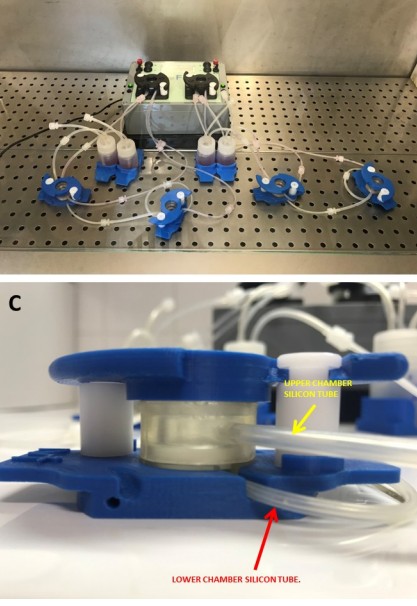
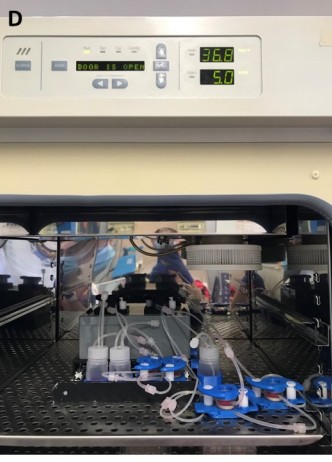

**Fig 1. The IVTech system settings.** A) Example of basic system size with two LB2, measuring stick in cm. B) The complete setting with four LB2 (for AngII vs control treatments) under laminar flow hood. C) Example of LB2 with outlet silicon tubes, in the figure the lower chamber is hidden by the blue support. D) The complete setting with four LB2 inside humidified incubator at 37˚C with 5% $CO_2$.

The hood surface was cleaned with the same detergents normally used to sterilize materials employed in cell cultures. The removable transparent glass bottom allows live imaging during culture and enables sample processing (Fig 2).

The possibility of having two independent circuits, one for the apical and one for the basal chamber, is an advantage that allows solving the problem of having two different media. Chambers are made in a biocompatible silicone polymer with self-sealing properties [4]. The chambers were connected by tubes, and the circuit dimensions were calculated using allometric laws [3]. The membrane is a polyester 0.45 μm-pore membrane, optically transparent and treated to permit cell adhesion; the pores allow the passage of cell metabolism products, preventing the translocation of the cells between the two environments of the bioreactor. Modules can be configured in different ways to obtain in series or in parallel circuits. The whole system is designed to be compatible with the most common laboratory instruments, like microscope and incubators, having the connected bioreactors the typical size of a multi-well plate (Fig 1A, 1B and 1D).

## Static cell cultures

Human neuroblastoma SH-SY5Y cells were obtained from ATCC (Manassas, VA) and cultured in T75 flasks in a humidified incubator at 37°C with 5% $CO_2$. SH-SY5Y cells were grown in Eagle's minimum essential medium (EMEM) supplemented with 10% fetal bovine serum, 1% penicillin–streptomycin, L-glutamine (2 mM), non-essential amino acids (1 mM), and sodium pyruvate (1 mM). HCASMC were obtained from Gibco and were cultured in a humidified incubator as SH-SY5Y cells (at 37°C with 5% $CO_2$). HCASMC were grown in Medium 231 supplemented with Smooth Muscle Growth Supplement and 1% penicillin–streptomycin. In MTT experiments, the cells were exposed to 1, 10 and 100 nM AngII (Sigma, A925) for 6, 24 and 48 hours. For Western blotting experiments, the cells were exposed to 10 nM AngII for 6 hours. The entire experiment was performed under a laminar flow hood.

## Dynamic cell cultures

HCASMC and SH-SY5Y cell types were seeded in two different LB2. The LB2 allows to monitor what happens in the first and second compartment independently, both for observations under the microscope (Fig 2), and for any sampling. In the connected setting, HCASMC were in contact with mediators eventually released by SH-SY5Y cells, thus simulating the crosstalk between tissues (Fig 3). SH-SY5Y cells were seeded at $2\times10^5$ cells/mL, on the glass in the bottom of LB2, with a tangential configuration, whereas HCASMC were seeded at $8\times10^4$ cells/mL

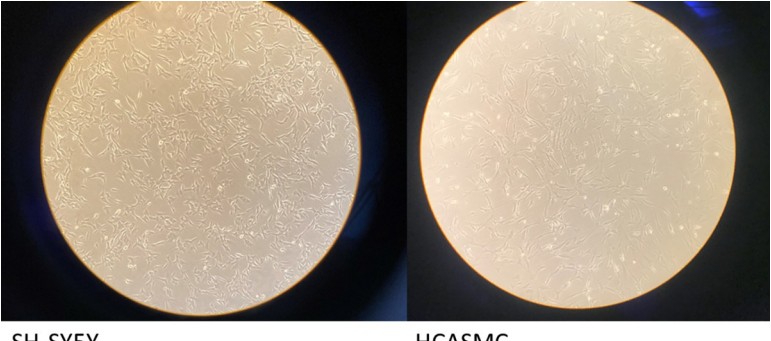

SH-SY5Y                                                    HCASMC

**Fig 2. SH-SY5Y cells (left) and HCASMCs (right) seeded under flow condition in two different LB2.** SH-SY5Y were seeded on the glass in the bottom of LB2, HCASMC were seeded onto the membrane within the other LB2.

onto the membrane within the other LB2 (Figs 2 and 3). LiveFlow® system together with the two LB2 were placed in the cell culture incubator at 37˚C with 5% CO2, to keep an aseptic condition (Fig 1D). Both cells lines were stimulated with 10 nM AngII. The entire experiment was performed under a laminar flow hood.

## Experimental design

The experimental design was developed according to the following steps:

1. Assessment of SH-SY5Y and HCASMC viability after AngII treatment in static conditions performed by MTT, after exposing for 6, 24 and 48 hours both cell types, seeded in petri dishes, to increasing concentrations of AngII (1-10-100nM).

2. Assessment of AngII-dependent PKCβII/HuR/VEGF activation in static conditions, in SH-SY5Y and HCASMC cell types, performed by Western blot.

3. Assessment of AngII-dependent PKCβII/HuR/VEGF activation evaluated, under dynamic conditions, in separately seeded SH-SY5Y and HCASMC cell types without any biohumoral exchange between them, and performed by Western blot.

4. Assessment of: a) AngII-dependent PKCβII/HuR/VEGF activation investigated, under dynamic conditions, in connected SH-SY5Y and HCASMC cell types, and performed by Western blot; b) VEGF release examined in SH-SY5Y and HCASMC respective medium evaluated by ELISA. Biohumoral exchange was possible through the porous membrane.

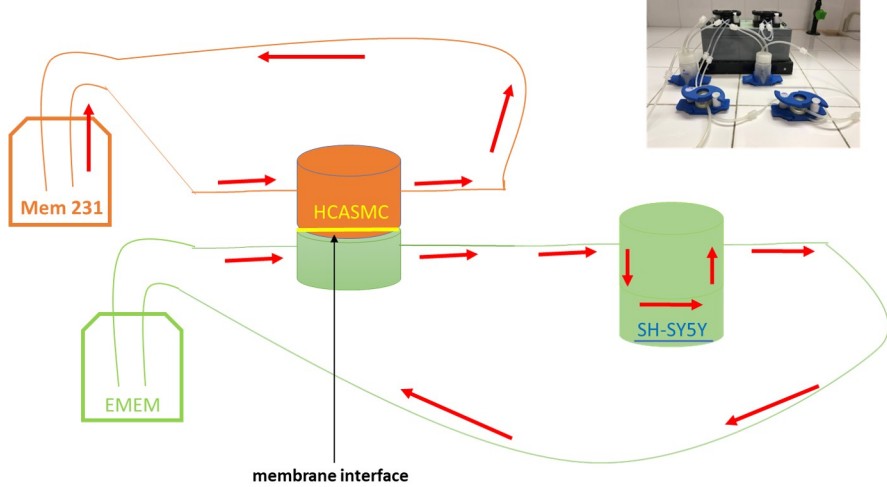

**Dynamic: HCASMC & SH-SY5Y connected through the membrane interface allowing humoral exchanges**

Mem 231

HCASMC

SH-SY5Y

EMEM

membrane interface

Flow direction

**Fig 3. The experimental setting enables to connect, under flow conditions, the two cell lines exposed to their own medium (orange: HCASMC; green: SH-SY5Y).** The mediators released from SH-SY5Y cells can interact with HCASMC cells through the membrane interface. This setting avoids the potential confounding effect of a common culture media, thus allowing the identification of specific factors released by each cell type. As indicated in Fig 1B, the bioreactor consists of two modules, one setting used for control and the other for treatment with AngII (10nM).

## MTT assay

Mitochondrial enzymatic activity was estimated by MTT [3-(4,5-dimethylthiazol-2-yl)-2,5-diphenyltetrazolium bromide] assay (Sigma). A cell suspension of $5x10^3$ cells/mL (for HCASMC cell line) and $2x10^4$ cells/mL (for SH-SY5Y cell line) was seeded into 96-well plates. Following each treatment of 6, 24 and 48 hours, 50 μL of MTT (concentration equal to 2.5 mg/mL) were added to each well. After incubation at 37˚C for 3 hours, the purple formazan crystals were formed. The formed crystals were solubilized in dimethylsulfoxide (DMSO; Sigma-Aldrich). Specifically, after removing the MTT from the wells, 100 μL of DMSO were added in order to lyse the cellular and mitochondrial membranes, and solubilize the formazan crystals. After 10 minutes, absorbance values were measured at 595 nm using a Synergy HT microplate reader (BioTek Instruments) and the results expressed as % with respect to control.

## Western blotting

Proteins were measured according to Bradford's method, using bovine albumin as internal standard. Proteins were diluted in 2xSDS protein gel loading solution, boiled for 5 min and separated on 12% SDS-PAGE. The anti-PKCβII rabbit polyclonal antibody (Santa Cruz), anti-HuR mouse monoclonal antibody (Santa Cruz) and the anti-VEGF rabbit monoclonal antibody (Abcam) were diluted based on each datasheet instructions. The nitrocellulose membrane signals were detected by chemiluminescence. The same membranes were re-probed with α-tubulin antibody and used to normalize the data. Statistical analysis of Western blot data was performed on the densitometric values obtained with the ImageJ image-processing program (https://imagej.nih.gov/ij).

## ELISA assay

The VEGF protein levels in SH-SY5Y and HCASMC cells were estimated in the respective medium with a specific ELISA kit (R&D Systems Inc.), according to the manufacturer's instructions.

This assay employs the quantitative sandwich enzyme immunoassay technique. A monoclonal antibody specific for VEGF is already pre-coated onto a microplate. Standards and samples were pipetted into the wells and any VEGF present was bound by the immobilized antibody. After washing away any unbound substances, an enzyme-linked polyclonal antibody specific for VEGF was added to the wells. Following a wash to remove any unbound antibody-enzyme reagent, a substrate solution was added and the colour developed in proportion to the amount of VEGF bound in the initial step. The colour development was stopped and the intensity of the colour measured (570/450 nm).

## Statistics

For statistical analysis the GraphPad Instat statistical package (GraphPad software, San Diego, CA, USA) was used. The data were analysed by analysis of variance (ANOVA) followed, when significant, by an appropriate *post hoc* comparison test, as detailed in the legends. Differences were considered statistically significant when p values ≤ 0.05.

# Results

## Static conditions

**Cell viability after AngII treatment.** The cell viability was studied after exposing both SH-SY5Y and HCASMC cell types for 6, 24 and 48 hours to increasing concentrations of AngII (1nM, 10nM and 100nM). SH-SY5Y cell viability was unaffected after 6 and 24 hours of

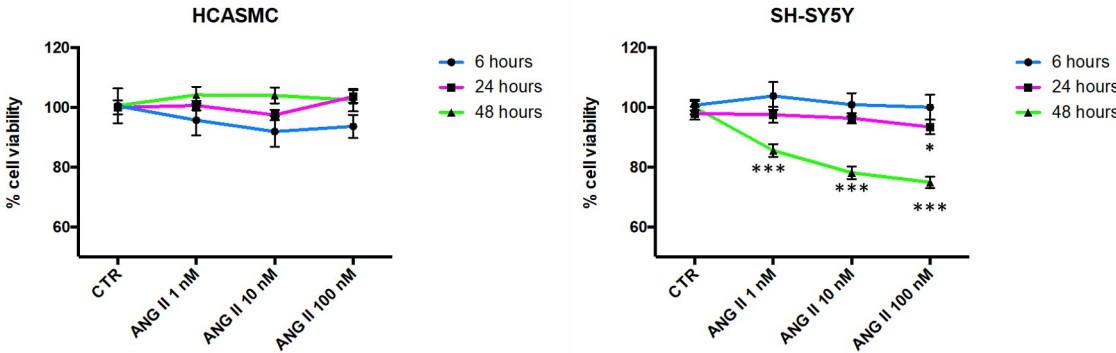

**Fig 4. Cell viability of SH-SY5Y cells after 6, 24- and 48-hours following Angiotensin II (ANG II) exposure at 1 nM, 10 nM, 100 nM.** The values are expressed as mean MTT [in %±S.E.M.]. ***p < 0.001, *p<0.05 vs control (CTR), Dunnett's multiple comparisons test (n = 16).

AngII exposure (at all the tested concentrations). Instead, after 48 hours treatment, consistent with previously published data [8], a significant decrease in SH-SY5Y viability was observed at all AngII concentrations (85.5%±2.1, 78.1%±2.1 and 74.9% ±1.9 at 1nM, 10nM and 100nM respectively). On the contrary, and as expected [9], MTT assay carried out on HCASMC did not show a significant decrease in cell viability following 6, 24 or 48 hours, at any AngII concentration (Fig 4).

Based on this evidence, the 10 nM AngII treatment for 6 hours was chosen, as it did not affect viability and it was well tolerated by both cell lines, and specifically by SH-SY5Y.

**AngII treatment and PKCβII/HuR/VEGF cascade.** There was no significant difference in PKCβII/HuR/VEGF cascade activation after 10 nM AngII treatments in both cellular types when seeded in static conditions (Fig 5).

## Dynamic conditions

**Parameters.** The following parameters have resulted to be the most suitable for cell growth in LB2:

a.  Cell density: 8x10^4 cells/mL for HCASMC and 2x10^5 cells/mL for SH-SY5Y.

b.  Flow Rate: 200 μL per minute in both circuits.

**AngII treatment and PKCβII/HuR/VEGF cascade in non-connected cells.** When cell types were separately seeded in dynamic conditions, the results confirmed what seen in static cultures, with no difference in PKCβII/HuR/VEGF cascade activation after AngII treatments in both cellular types (Fig 6).

**Cell lines in connection: AngII treatment and PKCβII/HuR/VEGF cascade.** Once connected each other as shown in Fig 3, following AngII exposure at 10 nM for 6 hours under flow conditions, there was a statistically significant increase of PKCβII/HuR/VEGF pathway activation in HCASMC (VEGF: +215.2% ±58.3 *vs* control, PKCβII: +85.8% ±34.5 *vs* control, HuR: +100.4% ±32.3 *vs* control), while no change was observed in SH-SY5Y cells (Fig 7). Nevertheless, we observed an AngII-dependent increase in VEGF protein release in SH-SY5Y medium (control: 80.5 pg/mL ± 8.8, AngII: 138.0 pg/mL ± 6.7, control vs treatment p = 0.002) while no changes were found in VEGF protein release in HCASMC medium (control: 93.5 pg/mL ± 33.5, AngII: 80.5 pg/mL ± 15.5, control vs treatment p = N.S.) (Fig 8).

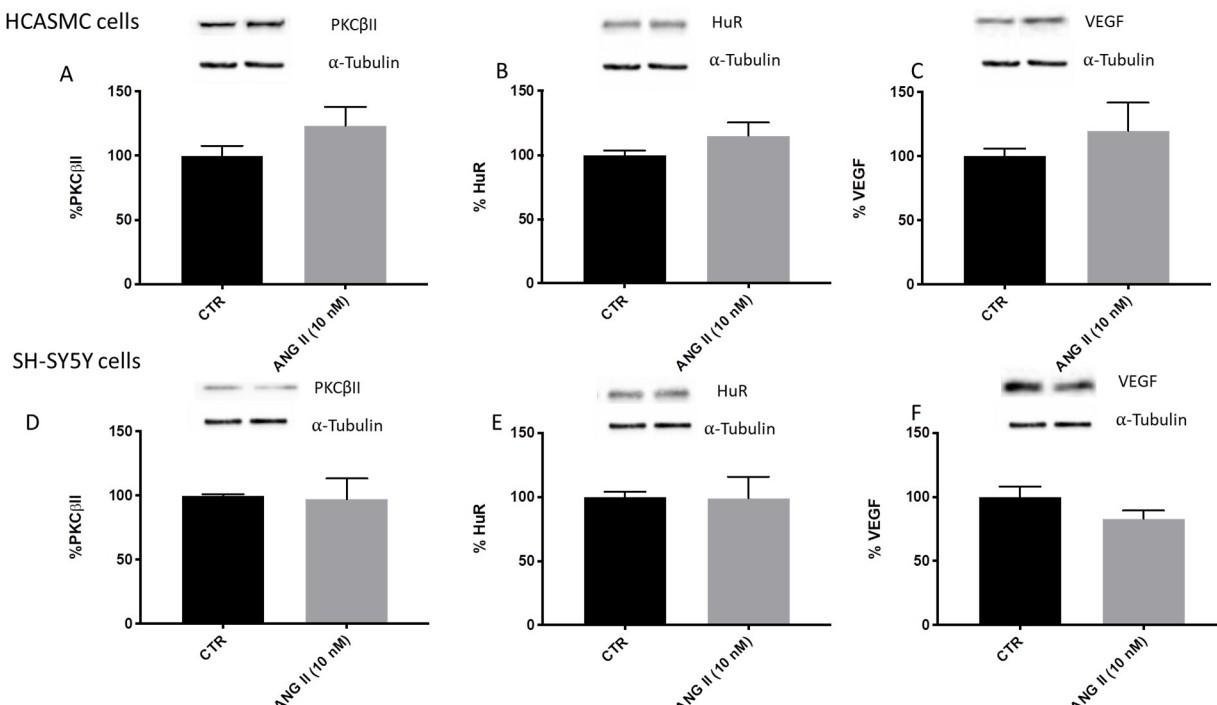

**Fig 5. Densitometric analysis and representative Western blotting of PKCβII, HuR and VEGF protein levels in the total homogenate of HCASMC (upper) and SH-SY5Y cells (lower) exposed to solvent (CTR) or Angiotensin II (ANG II) at 10 nM for 6 hours, in static condition.** PKCβII, HuR and VEGF bands were normalized to α-tubulin, and the results are expressed in percentage ± S.E.M. with respect to the control value (100%), n = 8–10.

## Discussion

Our group has been involved for years in studying the interactions between the nervous and cardiovascular systems [10–13], and the evidence that stress [11], pain [14] or peptides acting at central level [10, 15] might affect cardiovascular functions has pushed us to deepen the study of the crosstalk between the two systems. The main novelty of this study was to create an *in vitro* model able to explore the *in vitro* dialogue between two distinct human cell lines (SH-SY5Y; HCASMC), grown and connected in a bioreactor, representing a simplified model of the nervous-cardiovascular systems crosstalk.

The innovative feature of this work is represented by the possibility to have two different environments in LiveBox2, where the conditions can be set up by the user. In particular, two different environmental conditions were used: the apical side of the LiveBox2 chamber was filled with the SH-SY5Y medium, whereas the basal compartment was filled with the HCASMC medium. This is an innovative feature since, in general, a multi-organ approach needs to previously characterize the potential common medium, which has to be compatible with all the tissues within the pathway [16]. This process forces the cells to adapt to a novel environmental condition that is not the best option for them; therefore, this could cause a change in their behaviour. In this work, we solved the problem by using the native media developed for that specific cell type. Another innovative contribution consists in providing an experimental set up useful to culture cells belonging to different organs, connected under flow conditions and seeded in a dual flow bioreactor. Specific guidelines on flow rate and speed to use, or volume and type of culture medium were lacking and, in this regard, we have improved technical knowledge on bioreactors use for connecting cells growing in a different medium.

HCASMC cells: dynamic

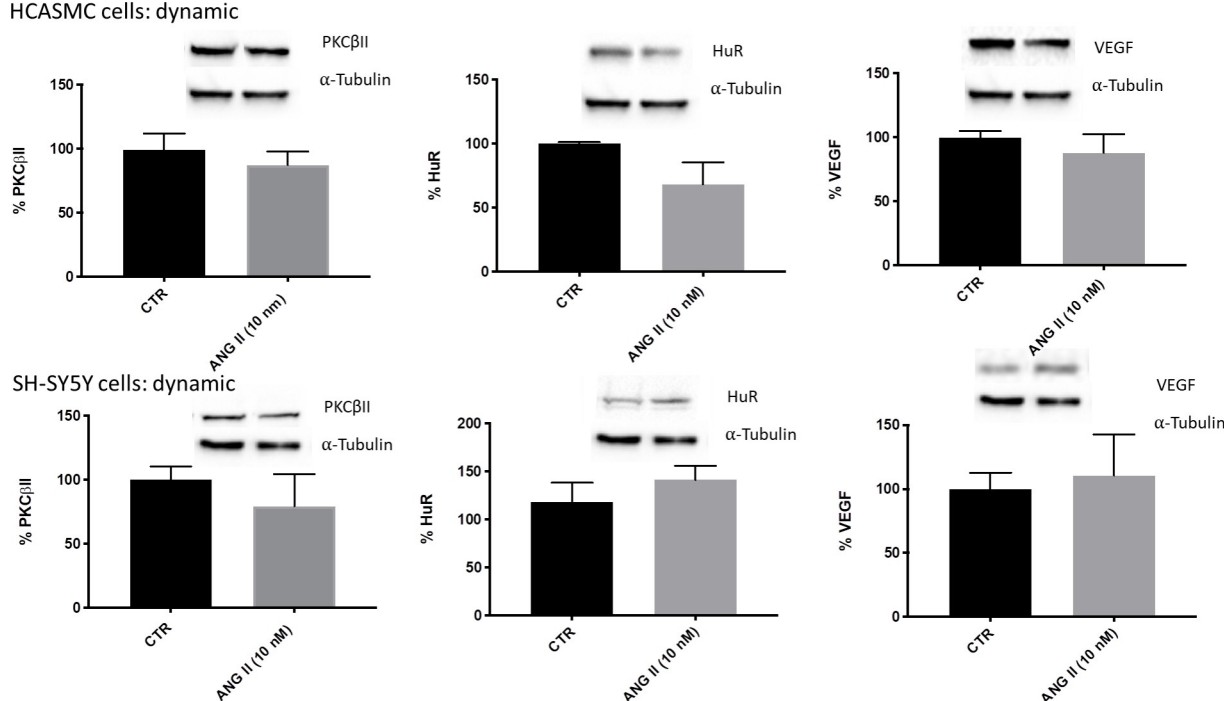

SH-SY5Y cells: dynamic

**Fig 6. Densitometric analysis and representative Western blotting of PKCβII, HuR and VEGF protein levels in the total homogenate of HCASMC (upper) and SH-SY5Y cells (lower) exposed to solvent (CTR) or Angiotensin II (ANG II) at 10 nM for 6 hours under flow conditions, not connected.** PKCβII, HuR and VEGF bands were normalized to α-tubulin, and the results are expressed in percentage ± S.E.M. with respect to the control value (100%), n = 3.

This also represented an improvement within the 3Rs research (Replacement, Reduction, Refinement of animal models), which develops new methods to enhance the quality standards of preclinical experimental models. To test the system, we explored AngII treatment effects on

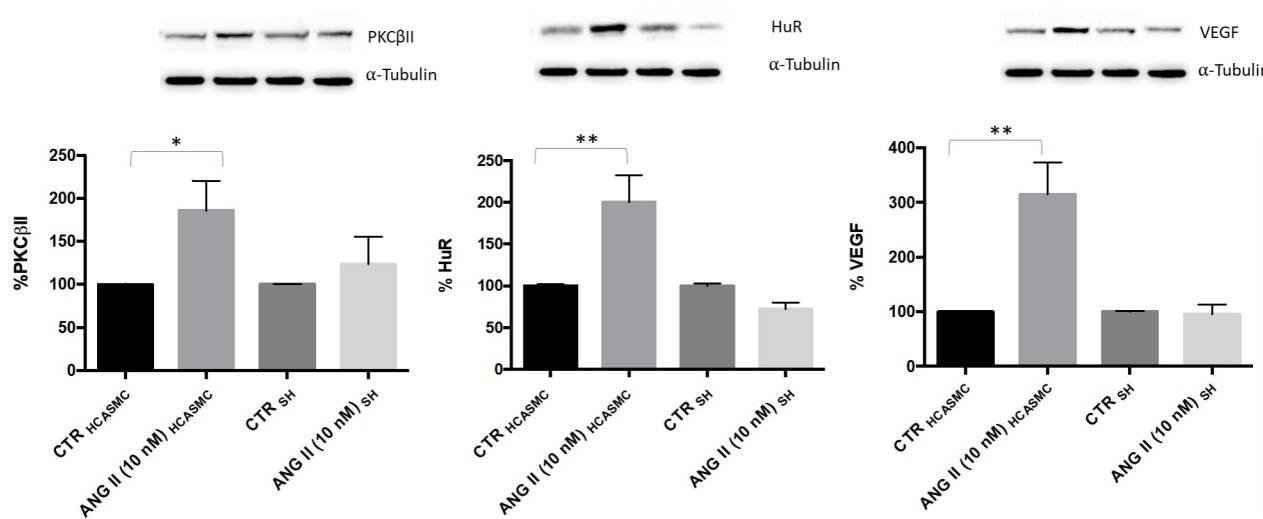

**Fig 7. Densitometric analysis and representative Western blotting of PKCβII, HuR and VEGF protein levels in the total homogenate from HCASMC and SH-SY5Y (SH) cells connected and exposed to solvent (CTR) or Angiotensin II (ANG II) at 10 nM under flow conditions for 6 hours.** Optical densities of PKCβII, HuR and VEGF bands were normalized to α-tubulin, and the results are expressed in percentage ± SEM with respect to the relative control value (100%). $^*p<0.05$; $^{**}p<0.01$; Unpaired t test, n = 5.

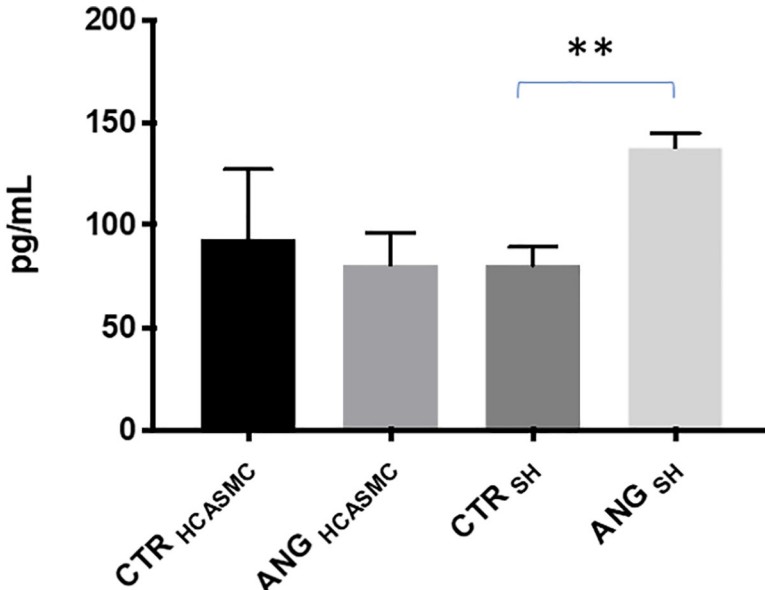

**Fig 8. Released VEGFA protein in the medium of cells, following treatment with Angiotensin II for 6 hours under flow conditions when cells were connected.** The release of VEGFA (pg/mL) was measured by ELISA and expressed as the means ± S.E.M. **p < 0.01, Student's t-test, n = 5.

the two cell types, seeded inside IVTech Bioreactor LiveBox 2 and connected under flow conditions, and we investigated AngII-dependent PKCβII/HuR/VEGF cascade activation in different experimental settings.

In mammal organisms, AngII has important roles in both cardiovascular and nervous systems: the renin-AngII system (RAS) constitutes one of the most important systems in the physiological regulation of blood pressure, and an inappropriate level of AngII is considered as a major risk factor in the development of cardiovascular diseases. Furthermore, several studies have demonstrated that an altered RAS may cause both neurodegeneration [17] and cardiovascular complications, often affecting each other' [18–21]. Moreover, although controversial [22], the existence of a so-called brain renin–angiotensin system [23–28], which might play a role in the regulation of neuroinflammation and progression to potential rehabilitation problems, is gaining interest, with concerns on societal and individual patient costs [11, 29–32]. Since Ang II is a potent stimulator of VEGF [33, 34] and its stimulus is able to favour the shuttling of HuR protein from the nucleus to the cytoplasm [35–37], a possible AngII-dependent activation of PKCβII/HuR/VEGF pathway was investigated in HCASMC and in SH-SY5Y, both in static and under flow conditions. The PKCβII/HuR/VEGF pathway is a molecular cascade, first described by our group in retinal bovine pericytes, which controls VEGF expression also under hypoxic conditions. In particular, PKCβII is able to increase VEGF protein expression through the RNA-binding protein ELAV/HuR [37–39].This *in vitro* model was settled through four main steps: first, in preliminary experiments, the cells under flow with respect to static conditions were observed, also selecting the more appropriate parameters for the dynamic setting. Second, AngII-dependent PKCβII/HuR/VEGF pathway activation in SH-SY5Y and HCASMC, both under static and dynamic conditions, were separately studied. Third, the best growing conditions to put the two cell lines in connection were identified. Finally, the two cell lines subjected to flow were connected and exposed to AngII treatment, evaluating the PKCβII/HuR/VEGF pathway in a condition of potential crosstalk.

We think we have reached the goal to culture and study different cell types in connection, mainly thanks to the use of a dual flow IVTech LiveBox 2 bioreactor system, which allows cells to grow in their own medium, enabling physio-pathological phenomena to be simulated *in vitro*. This system represents an innovative model, developed according to the 3Rs research objectives, that meets the necessary requirements of a bioreactor apparatus [40]. Finally, unlike the classical co-culture, dual-flow bioreactor made possible communication among different cell types, without the potential confounding effect of a common culture medium, thus allowing the identification of specific factors released by each cell type.

From the data collected during the setting of the system, and following the AngII treatment, we could make some preliminary considerations on cell response: according to literature [8], data on SH-SY5Y showed a weak decrease in viability (93.5%±1.8) after 24 hours of treatment with AngII 100 nM. MTT assay carried out on HCASMC did not show a significant decrease in cell viability following 6, 24 or 48 hours, at any AngII tested concentration. This is consistent with literature data showing that HCASMC treated at the same concentrations of angiotensin used in this work do not lose their ability to divide and migrate [9]. After the set-up of the system, we verified the feasibility of seeding and connecting HCASMC and SH-SY5Y cell types inside the dual flow model of IVTech LiveBox2 bioreactor under flow conditions, thus allowing biochemical communication between the two cell types, each grown in its culture medium. According to literature and to our preliminary experiments, a flow of 200 μL/min was applied in both circuits. In fact, in this model, higher flow rates caused cells detachment from the membranes and some evidence [41–43] suggested possible DNA damage following excessive shear stress. The behaviour displayed by the cells, once put in communication, has generated some critical information not otherwise collectable with previous methodologies:

1) In HCASMC connected to SH-SY5Y cells, AngII treatment caused an increased intracellular expression of VEGF through the activation of the PKCβII/HuR cascade, with no change in VEGF release in HCASMC medium. The absence of a change in VEGF in HCASMC medium, despite its increase in SH-SY5Y medium (Fig 8), indicates the good separation of the two media ensured by the membrane interface. However, the results suggest that VEGF released by SH-SY5Y was able to stimulate HCASMC laying onto the membrane, thus possibly activating the PKCβII/HuR cascade. Nevertheless, we cannot exclude that the duration of the experiment (6 hours) was not enough to allow HCASMC producing a measurable VEGF amount in the medium.

While VEGF is a pivotal factor for vascular development and angiogenesis [44] and its production is a well-known cell response to hypoxic conditions [45–50], HuR belongs to a small family of evolutionarily conserved RNA-binding proteins, named ELAV, which act at post-transcriptional level and are able to influence virtually any aspect of the post-synthesis fate of the targeted mRNAs [51]. Of interest, VEGF is a target of HuR and we previously demonstrated that, in the rat retina, diabetes-activated PKCβII/HuR cascade induces VEGF overexpression [37]. Further, the upstream inhibition of this cascade blunts these effects, thus supporting the concept that the PKCβII/HuR cascade can modulate, post-transcriptionally, VEGF expression through a route that is independent from the classic VEGF transcriptional control [37]. Notably, in hypoxic conditions, such as those also observed within the context of diabetic retinopathy, both the transcriptional and post-transcriptional pathways may be operant in controlling VEGF expression.

2) A significant increase of VEGF protein was observed in SH-SY5Y medium after AngII cell exposure and connection of the two cell lines.

The release of VEGF in the medium by SH-SY5Y cells may be interpreted as a reaction to hypoxia, this is in accordance with previously published data showing that oxygen deprivation caused VEGF production in SH-SY5Y at both mRNA and protein level, and with the evidence of a

correlation between AngII and hypoxic conditions [47, 48, 52–55]. SH-SY5Y cells have AngII type 1 receptor localized in the membrane [56], and when treated with AngII they displayed a stressed behaviour with subsequent upregulation of factors responsible for VEGF-mediated angiogenesis [8, 57]. Considering that we observed in HCASMC an increased intracellular VEGF expression only when HCASMC were connected with SH-SY5Y, it is tempting to speculate that a factor released by SH-SY5Y, perhaps VEGF itself [48], might have given to HCASMC a signal for a risk of hypoxia, thus promoting the activation of PKCβII/HuR/VEGF cascade in HCASMC.

Interestingly, the preliminary observations obtained after the connection of the two cell types through the bioreactor resemble more closely what seen *in vivo*. In particular, AngII administration in several *in vivo* models was reported to increase VEGF production, also via HuR activation, and to promote angiogenesis. In mice treated with Ang II infusion there was an increased VEGF expression inside the cells of the aortic wall [58]. In a model of cardiac hypertrophy induced by AngII administration, a significant increase in HuR cytoplasmic translocation, indicative of its activation, was observed in neonatal rat hypertrophic cardio-myocytes [59], thus emphasizing the involvement of HuR in the AngII-mediated increase of VEGF protein expression, as observed in our model.

AngII was also reported to enhance angiogenesis associated with tissue ischemia, via VEGF production. Indeed, in a murine model of myocardial ischemia [60], Ang II-pretreated rat mesenchymal stem cells showed enhanced VEGF synthesis, tube formation and angiogenesis *in vivo*, and in a model of femoral artery occlusion, AngII significantly increased VEGF protein content in ischemic hindlimb [61]. The same group described an AngII-dependent VEGF expression within the neovascular stromal interface in the Matrigel model in mice [62].

## Study limitation

This project was designed as a preliminary exploration of a system to culture cells belonging to different organs, connected under flow conditions and seeded in a dual flow bioreactor. The novelty of this work relies on the use of dual flow bioreactor system to culture and study the crosstalk between HCASMC and SH-SY5Y cell lines *in vitro*, each maintaining its own medium. Specific guidelines on flow rate and speed to use, or volume and type of culture medium were lacking [63] and, in this regard, we have improved technical knowledge on bio-reactors use for connecting cells growing in a different medium. Further applications by other investigators should validate the reliability and robustness of the system. Biochemical studies were used to test the system and not as main goal of our paper. Although we reported and dis-cussed some preliminary observations, a complete biochemical assessment of mediators released by each of the connected cells and a comprehensive study on their eventual prolifera-tion, migration and differentiation were not made. Nevertheless, considering the scope of this project, which was to test new technologies potentially useful in experimental medical research, we feel to have reached the goal of giving useful information on an innovative tool with a great potential in preclinical studies.

## Perspectives

Chronic disorders requiring rehabilitation therapies are difficult to investigate due to the extreme complexity of the interaction between the involved systems, e.g. nervous, cardiovascu-lar and/or immune. A comprehensive understanding of the crosstalk among them is still lacking and studies with appropriate methodological approaches are needed to explore in detail the reciprocal role of mediators and cell communication. In this context, this new model is able to be alongside the existing *in vivo*, *ex-vivo* and *in vitro* tools given its unique property of being simultaneously enough simple to allow specific observations and reasonably complex to more

closely reflect the physiological conditions of the cell's environment. This work does open new perspectives in future investigation on biochemical crosstalk between cells belonging to different organs and different systems, and describes a new model that can be combined with *in vivo* and classic *in vitro* models, supporting a global approach to 3R's in preclinical research.

## Acknowledgments

Authors are grateful to Dr. Lidia Cova for the precious suggestions.

## Author Contributions

**Conceptualization:** Alice Ghidoni, Alessia Pascale, Laura Calvillo.

**Data curation:** Nicoletta Marchesi, Annalisa Barbieri, Foroogh Fahmideh, Stefano Govoni, Alice Ghidoni, Gianfranco Parati, Emilio Vanoli, Alessia Pascale, Laura Calvillo.

**Formal analysis:** Nicoletta Marchesi, Annalisa Barbieri, Alessia Pascale.

**Funding acquisition:** Laura Calvillo.

**Investigation:** Nicoletta Marchesi, Annalisa Barbieri, Foroogh Fahmideh, Alessia Pascale, Laura Calvillo.

**Methodology:** Nicoletta Marchesi, Annalisa Barbieri, Foroogh Fahmideh, Alessia Pascale, Laura Calvillo.

**Project administration:** Laura Calvillo.

**Resources:** Alessia Pascale, Laura Calvillo.

**Supervision:** Stefano Govoni, Alessia Pascale, Laura Calvillo.

**Validation:** Alessia Pascale, Laura Calvillo.

**Visualization:** Laura Calvillo.

**Writing – original draft:** Laura Calvillo.

**Writing – review & editing:** Nicoletta Marchesi, Annalisa Barbieri, Stefano Govoni, Alice Ghidoni, Gianfranco Parati, Emilio Vanoli, Alessia Pascale, Laura Calvillo.

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
