## [Decision Letter · Decision Letter 0]

10 Aug 2020

PONE-D-20-22750

Use of dual-flow bioreactor to develop a simplified model of nervous-cardiovascular systems crosstalk

PLOS ONE

Dear Dr. Calvillo,

Thank you for submitting your manuscript to PLOS ONE. After careful consideration, we feel that it has merit but does not fully meet PLOS ONE’s publication criteria as it currently stands. Therefore, we invite you to submit an extensively revised version of the manuscript that addresses the major concerns raised during the review process.

We look forward to receiving your revised manuscript.

Kind regards,

Gaetano Santulli

Academic Editor

PLOS ONE

Journal Requirements:

Reviewers' comments:

Reviewer's Responses to Questions

**Comments to the Author**

1. Is the manuscript technically sound, and do the data support the conclusions?

Reviewer #1: Yes

Reviewer #2: Partly

2. Has the statistical analysis been performed appropriately and rigorously? 

Reviewer #1: Yes

Reviewer #2: N/A

3. Have the authors made all data underlying the findings in their manuscript fully available?

Reviewer #1: Yes

Reviewer #2: Yes

4. Is the manuscript presented in an intelligible fashion and written in standard English?

Reviewer #1: Yes

Reviewer #2: No

5. Review Comments to the Author

Reviewer #1: This manuscript uses a commercial, two chamber bioreactor to grow two different cell lines (a smooth muscle cell line and a neuroblastoma cell line) in vitro. Metabolites are exchanged between the two chambers so the two cell lines can chemically interact. Chemical interactions between the two cell lines have been studied extensively. The two chamber bioreactor is shown to allow chemical interactions in the manuscript. It is not clear to me, however, what is new and novel in this manuscript. The two chamber, commercial bioreactor has been demonstrated previously on other cell lines. Likewise, interactions between the cell lines studied here have been examine and measured previously. Starting on page 9, the manuscript identifies two, new 'critical' observations with this system: an increased expression of VEGF (this has been observed previously as referenced in the manuscript), and a significant increase of VEGF production. The manuscript does not offer "a more complete biochemical assessment" of the connection between these two cell lines, which would have been valuable. It is also difficult to evaluate the novelty of this manuscript because some of the authors are publishing a book (reference 7 in the manuscript) titled, "Brain and Heart Dynamics" that is not available until this fall.

Reviewer #2: Although the manuscript seemed to be written carefully, several crucial factors were missing owing to the nature of exp design. Therefore, the ms is far from mature at its current status.

1. How did the authors keeping the system in an aseptic condition?

2. Did the real crosstalk b/w nervous-cardiovascular really happened? More evidence should be provided.

3. Fig 1 and 2. Provide real photo with cells on the dish and/or container. We should know the growth condition of the two cell cultures.

6. PLOS authors have the option to publish the peer review history of their article (what does this mean?). If published, this will include your full peer review and any attached files.

Reviewer #1: No

Reviewer #2: No

---

## [Author Response · Author response to Decision Letter 0]

19 Oct 2020

1. Is the manuscript technically sound, and do the data support the conclusions?

Reviewer #1: Yes 

Reviewer #2: Partly

2. Has the statistical analysis been performed appropriately and rigorously?

Reviewer #1: Yes

Reviewer #2: N/A

3. Have the authors made all data underlying the findings in their manuscript fully available?

Reviewer #1: Yes

Reviewer #2: Yes

4. Is the manuscript presented in an intelligible fashion and written in standard English?

Reviewer #1: Yes

Reviewer #2: No

5. Review Comments to the Author

Reviewer #1: This manuscript uses a commercial, two chamber bioreactor to grow two different cell lines (a smooth muscle cell line and a neuroblastoma cell line) in vitro. Metabolites are exchanged between the two chambers so the two cell lines can chemically interact. Chemical interactions between the two cell lines have been studied extensively. The two chamber bioreactor is shown to allow chemical interactions in the manuscript. It is not clear to me, however, what is new and novel in this manuscript. The two chamber, commercial bioreactor has been demonstrated previously on other cell lines. Likewise, interactions between the cell lines studied here have been examine and measured previously. Starting on page 9, the manuscript identifies two, new 'critical' observations with this system: an increased expression of VEGF (this has been observed previously as referenced in the manuscript), and a significant increase of VEGF production. The manuscript does not offer "a more complete biochemical assessment" of the connection between these two cell lines, which would have been valuable. It is also difficult to evaluate the novelty of this manuscript because some of the authors are publishing a book (reference 7 in the manuscript) titled, "Brain and Heart Dynamics" that is not available until this fall.

We thank the Reviewer for these comments and for highlighting issues in need of better clarification in our paper. 

• In a section of the comments, the Reviewer wrote: “It is not clear to me, however, what is new and novel in this manuscript” and “The two chamber, commercial bioreactor has been demonstrated previously on other cell lines”

We thank the Reviewer for this relevant comment which allows us to improve the manuscript by clarifying an important point. From this comment we realized that the novelty of this work, that is the setup and use of dual flow bioreactor system to culture and study the crosstalk between HCASMC and SH-SY5Y cell lines in vitro, each grown in its own medium, was not described clearly enough. Therefore, we modified a few sentences in the text to better describe the contribution given by our work. In particular:

The innovative feature of LiveBox2 is represented by the possibility to have two different environments where the conditions can be set up by the user. In particular, we used two different environmental conditions: the apical side of the LiveBox2 chamber was filled with the SH-SY5Y medium, whereas the basal compartment was filled with HCASMC medium. As we have now specified in the new version of our paper, this is an innovative feature of the work since, in general, a multi-organ approach needs to previously characterize a potential common medium, which has to be compatible with all the tissues in the pathway [1]. This process forces the cells to adapt to a novel environmental condition that is not the best option for them; therefore, this could cause a change in their behaviour. In this work, we solved the problem by using the native media developed for that specific cell type.

Another innovative contribution of our paper consists in the description of an experimental set up useful to culture cells belonging to different organs, connected under flow conditions and seeded in a dual flow bioreactor. Specific guidelines on flow rate and speed to use, or volume and type of culture medium are not available, and, in this regard, we believe that with our work we have improved technical knowledge on bioreactors use for connecting cells growing in a different medium. 

The initial part of the discussion is now updated with a clearer statement about the innovative aspects of our work.

Regarding the comment on commercial bioreactors, it is true that the use of bioreactors has already been validated for the study of several cell lines, including neuroblastoma cells or HCASMC. Nevertheless, the studies available in PubMed after searching the keywords: “neuroblastoma AND bioreactor”-“bioreactor AND coronary artery”- “human coronary artery smooth muscle cell AND bioreactor”, gave no information regarding a crosstalk between the two cell lines seeded in co-culture, under flow conditions. Considering the importance of the reviewer comment, we briefly report here below the most interesting papers published in this field. 

A co-culture under flow of neuroblastoma SK-N-BE(2) with HUVECcells is reported by Villasante et al (Vascularized Tissue-Engineered Model for Studying Drug Resistance in Neuroblastoma Theranostics. 2017; 7(17): 4099–4117. doi: 10.7150/thno.20730), who investigated the effects of retinoid therapy on tumor vasculature and drug-resistance. 

Izzo et al. (Biomed Microdevices. 2019; 21(1): 29. Influence of the static magnetic field on cell response in a miniaturized optically accessible bioreactor for 3D cell culture) investigated the influence of the static magnetic field on SH-SY5Y neuroblastoma cells response in a bioreactor for 3D cell culture, measuring heat-shock protein 70 (Hsp-70), Bcl-2 and Bax. 

An interesting work (Yahya Elsayed et al. Modeling, simulations, and optimization of smooth muscle cell tissue engineering for the production of vascular grafts. Biotechnol Bioeng 2019 Jun;116(6):1509-1522) describes a smooth muscle cell tissue engineering, useful for the production of vascular grafts, by using a scaffold in a dynamic bioreactor with a rotating shaft. Mechanical properties and oxygenation level were studied, without investigating metabolic pathways. Again, no co-culture was present.

Sharifpoor and colleagues (Functional characterization of human coronary artery smooth muscle cells under cyclic mechanical strain in a degradable polyurethane scaffold -Biomaterials. 2011 Jul;32(21):4816-29) describe a degradable polyurethane scaffold to grow HCASMC, useful for eventual use in bioreactor.

Balcells et al. (Smooth muscle cells orchestrate the endothelial cell response to flow and injury Circulation. 2010 May 25;121(20):2192-9) exposed a co-culture of smooth muscle cells and endothelial cells to coronary artery flow in a perfusion bioreactor, but they studied the mTOR pathway and the expression of phospho-S6 ribosomal protein in both cells.

Kural and colleagues described an in vitro human arterial injury model in bioreactor, helpful in the study of smooth muscle cells and endothelial cells interactions, without any involvement of neuroblastoma cell line.

As clearly shown by the above list of published works, there are no studies using dual-flow bioreactors, and specifically IVTech bioreactors, which are focussed on a co-culture of SH-SY5Y cells and HCASMC, and their crosstalk. In this regard, our work offers novel information, also considering the fact that this system allows two different cell types to grow and to communicate while being cultured in their own medium, thus allowing the identification of specific factors released by each cell type without the potential confounding effect of a common culture medium.

• In another section, the Reviewer wrote: Likewise, interactions between the cell lines studied here have been examined and measured previously. Starting on page 9, the manuscript identifies two, new 'critical' observations with this system: an increased expression of VEGF (this has been observed previously as referenced in the manuscript), and a significant increase of VEGF production.

These cell lines were previously extensively studied and mentioned in several papers, however, they have never been studied in connection under flow condition, and in the context of a bioreactor tool. The co-culture of these cell lines in the bioreactor, and exploration of their crosstalk, are completely novel. 

Considering this important point of criticism, we have repeated the literature search, and we provide here below a summary of the results of such a search. When we searched in PubMed the combined keywords “SH-SY5Y and HCASMC” and “SH-SY5Y and coronary artery”, we were not able to find papers reporting a complete pattern of biochemical interactions between the two lines, nor methodological works describing a SH-SY5Y and HCASMC co-culture in a dual flow bioreactor.

Below, we briefly report the most interesting papers published in this field among which, as mentioned above, there are no studies investigating in depth the crosstalk between SH-SY5Y and HCASMC. Moreover, there are no methodological works describing a SH-SY5Y and HCASMC co-culture system in a dual flow bioreactor.

The following paper:

Srivastava, S., Blower, P.J., Aubdool, A.A.et al.Cardioprotective effects of Cu(II)ATSM in human vascular smooth muscle cells and cardiomyocytes mediated by Nrf2 and DJ-1.Sci Rep6,7 (2016). https://doi.org/10.1038/s41598-016-0012-5

where HCASMC are used, cite neuroblastoma cells only as reference to state that “Excess intracellular Cu(II) levels are known to cause mitochondrial toxicity and dysfunction” in SH-SY5Y.

A published thesis partly available at https://search.proquest.com/openview/82d8708084462075a51e84c9dfafd7f7/1?cbl=18750&diss=y&pq-origsite=gscholar

investigated the “effects of Mg ions on human coronary artery smooth muscle cells (HCASMCs)” in a “In vitro Biocompatibility Evaluation of Biodegradable Metals for Cardiovascular Stent Application”. In this work, SH-SY5Y cells are cited as test used for assessment of in vitro corrosion and cytotoxicity of a magnesium alloy.

When the keywords inserted in PubMed were: “human coronary artery smooth muscle cell AND neuroblastoma”, the only result was J B Smith et al. Lowering extracellular pH evokes inositol polyphosphate formation and calcium mobilization J Biol Chem. 1989 May 25;264(15):8723-8

where authors found that “a decrease in pH, triggers cell Ca2+ mobilization in fibroblasts, endothelial, smooth muscle, and neuroblastoma cells.

When the keywords inserted in PubMed were: “SH-SY5Y AND coronary artery”, the result was Shi et al. Panax notoginseng saponins provide neuroprotection by regulating NgR1/RhoA/ROCK2 pathway expression, in vitro and in vivo. J Ethnopharmacol. 2016 Aug 22; 190:301-12. doi: 10.1016/j.jep.2016.06.017. Epub 2016 Jun 8.

In this paper, the investigators showed that Panax notoginseng saponins provided neuroprotective effects in a rat model of cerebral ischemia and in SH-SY5Y cells exposed to oxygen/glucose deprivation injury. The term “coronary” is related to the used of Panax notoginseng in the context of coronary heart disease.

In the work by Sugimoto et al. (Neuroblastoma Cell Lines Showing Smooth Muscle Cell Phenotypes-Diagnostic Molecular Pathology: December 2000 - Volume 9 - Issue 4 - pp 221-228), authors found that “S-type cells have either the immature or mature smooth muscle cell phenotype, and neural crest cells very likely have the ability of to differentiate into smooth muscle cells in the human system”. In this paper, the expressions of smooth muscle-specific proteins were described in several neuroblastoma parents and clones, and no chemical interactions between cells were described.

When the keywords “human coronary artery smooth muscle cells AND neuroblastoma” were inserted in Google Scholar, the reported papers described the behaviour of either cell separately, citing the other cell line as reference, e.g. as test used for various assessments. We were not able to find works describing co-culture of HCASMC and SH-SY5Y cells, neither in classic petri dishes nor under flow conditions.

In case the Reviewer is aware of other papers on these issues that we might have missed, we would be grateful if he/she could share them with us, so that we might refer to them in relation to our findings.

Finally, the increase of VEGF expression in the context of AngII infusion has been previously reported, and has been replicated in different and heterogeneous experimental settings. We used and cited these published data to reinforce our results obtained in the bioreactor system. 

• The Reviewer commented that “The manuscript does not offer "a more complete biochemical assessment" of the connection between these two cell lines, which would have been valuable”. 

We are grateful for this stimulating suggestion. Following this comment, we have performed a further experiment aimed indeed at improving the biochemical assessment. We measured VEGF release in the medium of HCASMC to have more biochemical information. This new result has been integrated in the manuscript (results and discussion sections), and a new figure describing these results is now added (the new figure 8).

As the Reviewer can notice, our paper does not provide a complete biochemical assessment of the co-culture, that is beyond the scope of the work. Biochemical studies were used to test the system and their performance was not the main aim of our study. This methodological issue is now clearly stated in the study limitations section. 

• Finally, the Reviewer noted that the publication "Brain and Heart Dynamics" is not available until this fall. 

We thank the Reviewer for reporting this problem. A copy of the cited book chapter is now provided for the Reviewer as attachment.

Reviewer #2: Although the manuscript seemed to be written carefully, several crucial factors were missing owing to the nature of exp design. Therefore, the ms is far from mature at its current status. 

The Reviewer made important comments and we thank him/her for the opportunity of pointing out some critical aspects, thus offering us the possibility to better specify the main goal of this research work and to improve its presentation. 

Since bioreactor technology is relatively recent, information on a number of basic aspects is still lacking (e.g. culture conditions, grown condition for several type of cells, flow parameters etc.). Moreover, with respect to the classic in vitro models, extensively used within the last decades, for which several procedures are available, only a few tried-and-tested experimental protocols are available for researchers to work under dynamic conditions, especially when considering specific interconnected cell lines. Therefore, our primary endpoint was to develop and share with the scientific community a new co-culture set-up involving specifically SH-SY5Y cells and HCASMC, where each cell type is seeded in its own medium and under flow conditions, in a more physiological environment. Therefore, the experimental design was developed accordingly, giving priority to technical information over biomedical information. Our idea was to accelerate the diffusion of these innovative experimental models, allowing others to more deeply study the topics we approached. Moreover, our work is characterized by an innovative feature: usually a multi-organ approach needs to previously characterize the potential common medium that has to be compatible with all the cells involved [1]. This process forces the cells to adapt to a different medium, potentially causing changes in their behaviour. In this work, we solved this problem by using the native medium developed for that specific cell type. These aspects are now described in the text, in the introduction and discussion sections.

1. How did the authors keeping the system in an aseptic condition? 

We thank the Reviewer for this comment which allows us to add more details to our material and methods paragraph. All the components of the IVTech bioreactor are autoclaved and the entire experiment is performed under a laminar flow hood. The surface is cleaned with the same detergents normally used to sterilize materials used in cell cultures. Membranes were conditioned keeping them in ethanol 70% for two hours and exposed to UV light for 15 min, before cell seeding. Figure 1 describing this setting has been improved, and the material and methods section has been updated accordingly.

2. Did the real crosstalk b/w nervous-cardiovascular really happened? More evidence should be provided. 

To better describe the crosstalk between nervous/cardiovascular cells, we have performed a further experiment, measuring VEGF release in the medium of HCASMC. These new data are now added in the text and illustrated in an additional figure (Figure 8). As stated in the paper, the main endpoint of this work was to test new technologies potentially useful in preclinical research, and to present reliable setup data on the co-culture of HCASMC and SH-SY5Y cell lines in a dual flow bioreactor.

Admittedly, all the observations/results we reported on the nervous/cardiovascular crosstalk should be considered exploratory data obtained with the use of this novel system. Our findings would now need to be confirmed and validated by additional studies. A complete biochemical assessment was not provided since was beyond the scope of this work. This point is now more clearly addressed in the “Study limitation” section.

3. Fig 1 and 2. Provide real photo with cells on the dish and/or container. We should know the growth condition of the two cell cultures. 

We thank the Reviewer for this suggestion. A new figure (Figure 2) including a real photo with cells on the container and a short sentence giving more information on the growth condition of the two cell cultures are now provided.

Bibliography

1. Ahluwalia A, Misto A, Vozzi F, Magliaro C, Mattei G, Marescotti MC, et al. Systemic and vascular inflammation in an in-vitro model of central obesity. PLoS One. 2018;13(2):1–15.

---

## [Decision Letter · Decision Letter 1]

4 Nov 2020

PONE-D-20-22750R1

Use of dual-flow bioreactor to develop a simplified model of nervous-cardiovascular systems crosstalk:a preliminary assessment

PLOS ONE

Dear Dr. Calvillo,

Thank you for submitting your manuscript to PLOS ONE. After careful consideration, we feel that it has merit but does not fully meet PLOS ONE’s publication criteria as it currently stands. Therefore, we invite you to submit a revised version of the manuscript that addresses the points raised during the review process.

We look forward to receiving your revised manuscript.

Kind regards,

Gaetano Santulli, MD

Academic Editor

PLOS ONE

Reviewers' comments:

Reviewer's Responses to Questions

**Comments to the Author**

1. If the authors have adequately addressed your comments raised in a previous round of review and you feel that this manuscript is now acceptable for publication, you may indicate that here to bypass the “Comments to the Author” section, enter your conflict of interest statement in the “Confidential to Editor” section, and submit your "Accept" recommendation.

Reviewer #2: All comments have been addressed

2. Is the manuscript technically sound, and do the data support the conclusions?

Reviewer #2: Yes

3. Has the statistical analysis been performed appropriately and rigorously? 

Reviewer #2: Yes

4. Have the authors made all data underlying the findings in their manuscript fully available?

Reviewer #2: Yes

5. Is the manuscript presented in an intelligible fashion and written in standard English?

Reviewer #2: Yes

6. Review Comments to the Author

Reviewer #2: I am happy with the revision and only two minors should be revised.

Please improve the quality of Fig 3.

Please explain how the reactor was kept aseptic.

7. PLOS authors have the option to publish the peer review history of their article (what does this mean?). If published, this will include your full peer review and any attached files.

Reviewer #2: No

---

## [Author Response · Author response to Decision Letter 1]

5 Nov 2020

Response to Reviewers

Comments to the Author

1. If the authors have adequately addressed your comments raised in a previous round of review and you feel that this manuscript is now acceptable for publication, you may indicate that here to bypass the “Comments to the Author” section, enter your conflict of interest statement in the “Confidential to Editor” section, and submit your "Accept" recommendation.

Reviewer #2: All comments have been addressed

We thank the Reviewer for the time dedicated to our work

2. Is the manuscript technically sound, and do the data support the conclusions?

Reviewer #2: Yes

3. Has the statistical analysis been performed appropriately and rigorously?

Reviewer #2: Yes

4. Have the authors made all data underlying the findings in their manuscript fully available?

Reviewer #2: Yes

5. Is the manuscript presented in an intelligible fashion and written in standard English?

Reviewer #2: Yes

6. Review Comments to the Author

Reviewer #2: I am happy with the revision and only two minors should be revised.

We thank the Reviewer for her/his approval

Please improve the quality of Fig 3.

We thank the Reviewer for her/his comment. We have improved the quality of Fig. 3 by uploading the file to the Preflight Analysis and Conversion Engine (PACE) digital diagnostic tool (https://pacev2.apexcovantage.com/). PACE helps ensure that figures meet PLOS requirements. It is possible to download the corrected figure by following the link in the pdf sent for approval.

Please explain how the reactor was kept aseptic.

Thank you for this suggestion. We report the complete sterilizing techniques protocol:

Cell Culture & IVTECH Bioreactor

The staff who worked on this project always followed all the good standard procedures:

• wash hands thoroughly before and after working with cell cultures;

• wear a lab coat, gloves and safety glasses to protect themselves from any hazardous materials and to prevent contamination of the cell cultures from microbes present on skin and clothes.

Work area 

• Static and dynamic cell cultures (IVTECH Bioreactor) were carried out in a biosafety cabinet, under a laminar flow, located in an ad hoc cell culture room.

• The inside of the biosafety cabinet was wiped with disinfectant (benzalkonium chloride solution) followed by 70% ethanol before and after use. Further, UV light was turned on to sterilize the biosafety cabinet when unused.

• Areas of work only contained the items required for the current procedures. 

Handling & sterilizing techniques

• 70% ethanol was sprayed on gloved hands before starting the work in the biosafety cabinet.

• All the equipment and reagents (cell culture media or other reagents) utilized into the biosafety cabinet were sprayed beforehand with 70% ethanol.

• Disposable plastic pipettes were used to manipulate solutions, Pasteur glass pipettes were autoclaved as well as each single component of the bioreactor (connector, holder, upper and lower cylinders, top and bottom cylinders, thin glasses reservoirs). Specifically, each component was wrapped in aluminum foil and, after autoclaving, all the packages were opened into the hood and the bioreactor circuit was assembled in the hood itself under laminar flow.

• The sterilization of the polyester membranes was achieved by keeping them in ethanol 70% for two hours followed by UV light exposure for 15 min.

---

## [Editor Report · Decision Letter 2]

6 Nov 2020

Use of dual-flow bioreactor to develop a simplified model of nervous-cardiovascular systems crosstalk:a preliminary assessment

PONE-D-20-22750R2

Dear Dr. Calvillo,

We’re pleased to inform you that your manuscript has been judged scientifically suitable for publication and will be formally accepted for publication once it meets all outstanding technical requirements.

Kind regards,

Gaetano Santulli, MD

Academic Editor

PLOS ONE

---

## [Editor Report · Acceptance letter]

12 Nov 2020

PONE-D-20-22750R2 

Use of dual-flow bioreactor to develop a simplified model of nervous-cardiovascular systems crosstalk: a preliminary assessment. 

Dear Dr. Calvillo:

I'm pleased to inform you that your manuscript has been deemed suitable for publication in PLOS ONE. Congratulations! Your manuscript is now with our production department. 

Kind regards, 

on behalf of

Prof. Gaetano Santulli 

Academic Editor

PLOS ONE